# A Pilot Study Comparing a Micronized Adipose Tissue Niche versus Standard Wound Care for Treatment of Neuropathic Diabetic Foot Ulcers

**DOI:** 10.3390/jcm11195887

**Published:** 2022-10-05

**Authors:** Sik Namgoong, In-Jae Yoon, Seung-Kyu Han, Ji-Won Son, Jeehee Kim

**Affiliations:** 1Department of Plastic Surgery, Korea University College of Medicine, Seoul 08308, Korea; 2Diabetic Wound Center, Korea University Guro Hospital, Seoul 08308, Korea; 3R&D Center, ROKIT Healthcare, Seoul 08514, Korea

**Keywords:** diabetic foot, micronized adipose tissue

## Abstract

Numerous studies have demonstrated the various properties of micronized adipose tissue (MAT), including angiogenic, anti-inflammatory, and regenerative activities, which can be helpful in wound healing. This exploratory clinical trial aimed to report the efficacy and safety of MAT niche for treating diabetic foot ulcers. Twenty subjects were randomly divided into MAT niche treatment (*n* = 10) and control groups (*n* = 10). All patients were followed up weekly for 16 weeks. We evaluated the efficacy of the MAT niche treatment by assessing the (1) reduction in wound area after 4 weeks and (2) percentage of patients who achieved complete wound closure after 16 weeks. All possible adverse events were recorded. The wound area was reduced by 4.3 ± 1.0 cm^2^ in the treatment group and by 2.0 ± 1.1 cm^2^ in the control group (*p* = 0.043). Complete wound healing was achieved after 16 weeks in eight out of 10 patients (80%) in the treatment group and three out of six (50%) in the control group (*p* = 0.299). No serious adverse events related to MAT niche treatment were observed. Although the present study’s findings do not support the use of this therapy to treat foot ulcers of patients with diabetes owing to the small number of patients included and the absence of statistical significance, the results of this pilot preliminary study are promising in that MAT niche autografts may offer the possibility of a simple and effective treatment for diabetic ulcers. Further follow-up studies with a larger number of patients are required to validate our findings.

## 1. Introduction

Since the first report on autologous adipose tissue graft was published in the early 20th century [1], it has long been commonly used in cosmetic and reconstructive surgery [2]. Initially, adipose tissue grafts were used for their volume-increasing effect, such as in breast reconstruction secondary to oncologic resection or facial volumizing secondary to age-related volume loss. However, adipose-derived stem cells (ASCs) were discovered by Zuk et al. in 2002 [3], promoting a plethora of research on the regenerative properties of adipose tissue. Thus, the scope of clinical application of adipose tissue graft is now being expanded beyond volumizing procedures to skin rejuvenation procedures [4,5,6] and treatment of wounds [7,8,9], among others. 

Recently, micronized adipose tissue (MAT) obtained by mechanical dissolving, as opposed to collagenase usage, has been newly developed and has demonstrated positive effects of angiogenesis, antioxidant properties, and protein synthesis in vitro [10] and in vivo [11]. MAT has been demonstrated to have favorable therapeutic effects in treating scars and improving wrinkles clinically [12]. Considering the regenerative potential of MAT, which is composed of (1) cellular components, such as ASCs, fibroblasts, endothelial progenitor cells, and immune cells, (2) extracellular matrices (ECMs), and (3) cytokines, we hypothesized that MAT grafting may be a promising alternative for diabetic foot ulcer (DFU) treatment. However, to the best of our knowledge, no clinical studies have reported the effects of MAT on wound healing in patients with diabetes, who commonly have compromised healing processes. This pilot study aimed to compare autologous MAT grafting with conventional treatment, predominantly targeting the efficacy of improving the wound healing rate in DFUs.

## 2. Materials and Methods

This study protocol was approved by the Institutional Review Board of the authors’ institution (No. 2020GR0095). The trial was performed (enrollment, treatment, clinical assessments, and result analyses) between March 2020 and October 2020 in the Diabetic Wound Center and the Department of Plastic Surgery of the Korea University Guro Hospital, Seoul, Republic of Korea. This study was conducted in accordance with the Declaration of Helsinki, and informed consent was obtained from each patient.

### 2.1. Patients

Patients were selected according to the following main inclusion criteria: age > 19 years, having diabetes mellitus for >5 years, presence of lower-extremity ulcers that did not show signs of healing (wound epithelialization, granulation formation, and wound contraction) in the 6 weeks prior to inclusion, and transcutaneous oxygen pressure > 30 mmHg or the presence of a foot pulse detected using Doppler sonography at the wound location during the screening process. Patients who were capable and willing to provide consent and agreed to comply with the study procedures and follow-up evaluations were included. Patients with infection, cellulitis, or osteomyelitis diagnosed by infection signs, magnetic resonance imaging, or microbiologic culture results were excluded. Patients with a history of participating in another clinical study within 4 weeks, those diagnosed with malignant tumors or a concurrent illness that might interfere with wound healing (e.g., immunocompromised, treated with corticosteroids or chemotherapy, systemic infectious state, or connective tissue disorders), or those unable to tolerate offloading were also excluded. All ulcers were subjected to sharp debridement to remove necrotic or hyperkeratinized tissue and control infection prior to study initiation. 

Eligible patients were enrolled and randomized into either of the treatment groups (MAT niche treatment group, test group vs. standard dressing care, control group). Randomization schedules were stratified using a permuted block method with a random block size and a statistical analysis system and treatment allocation ratio of 1:1 (Figure 1).

Twenty patients with DFUs, comprising 17 men and three women with a mean age of 58.1 ± 10.5 (range, 34–71) years, were included. Among the 20 patients, 10 each were assigned to the treatment and control groups. In the primary evaluation period, two patients in the control group were excluded; one patient dropped out because of a medical condition (hypoglycemia), and the other chose to withdraw. The secondary evaluation period, which was conducted only for patients who could be followed up until 16 weeks post treatment, included 16 patients. Therefore, an intention-to-treat analysis was conducted on the data available from the 18 patients, who were treated during the primary evaluation period. The relatively long-term follow-up data available from the 16 patients were analyzed for the secondary efficacy criterion (Figure 1). Baseline patient information of the two groups is shown in Table 1. No significant differences were observed regarding any clinical characteristics between the groups.

#### 2.1.1. Brief Management Protocol of Diabetic Foot Ulcers in Our Diabetic Wound Center

A complete medical history was obtained from each patient. General serologic tests, including those for blood glucose and other inflammatory markers, were performed. To evaluate the vascularity of diabetic foot, transcutaneous partial oxygen tension, Doppler wave, and toe pressure were measured. Patients with peripheral arterial disease received percutaneous transluminal angioplasty from an interventional cardiologist. For the management of wound bioburden, deep tissue culture was performed. When necessary, intravenous antibiotics were administered empirically and were then changed according to the results of culture and sensitivity tests. To evaluate neuropathy, a Semmes—Weinstein monofilament test, pin prick test, temperature test, electromyography, and nerve conduction velocity test were conducted. Appropriate offloadings were provided according to ulcer locations. All the abovementioned management protocols were routinely applied to all patients who visited our diabetic wound center for treatment of diabetic ulcers. Appropriate patients without infection who met the study’s inclusion criteria were enrolled in the present clinical trial. 

The ulcers which need debridement were debrided at each visit before the enrollment of the present study in both groups. Debridement was performed on the wound bed until a healthy bleeding base was reached. After the initiation of the present study, the ulcers did not undergo debridement in both groups.

#### 2.1.2. Treatment Group: MAT Niche Creation

Thirty to fifty milliliters of abdominal adipose tissue was obtained from the patients by liposuction using a cannula with a 3 mm inner diameter under local anesthesia. The aspirated adipose tissue was immediately micronized mechanically with a micronizer filter (Adinizer; BSLrest, Busan, Korea) in descending order of filter pore sizes to reduce the particle size of the fat tissue from 2400 μm to 1200 μm to 600 μm to 200 μm. 

The processed micronized fat was collected in a syringe to mix with saline. Normal saline was then added to the collected micronized fat and gently inverted up and down for the washing purification process. The micronized fat filled with normal saline solution was separated by layers in a vertical syringe condition a few minutes later by gravity. The upper layer was retained; the lower layer containing oily substances and blood residues was removed. The retained yellowish upper layer was the final MAT product.

Three-dimensional (3D) image files were created on the basis of wound photographs taken in the test group and tailored to each patient’s respective wound. All stereolithography files were processed using the NewCreatorK (ROKIT Healthcare, Seoul, Korea) software and sliced into 1 to 5 mm thick layers to generate G-code instructions for the 3D Bioprinter (Dr. INVIVO, ROKIT Healthcare). G-code instruction sets were sent to the printer using the NewCreatorK (ROKIT Healthcare), a 3D Bioprinter host program. After uploading the G-code file to the 3D bioprinter, a scaffold was firstly printed with medical-grade polycaprolactone (PCL, molecular weight = 45,000; Evonik Nutrition & Care GmbH, Essen, Germany) using the air dispenser of the 3D bioprinter at 100–120 °C and 500–800 kPa.

After printing the PCL scaffold, the MAT-filled syringe (dispenser I) and fibrin glue (Tisseel; Baxter AG, Vienna, Austria)-filled syringe (dispenser II) were inserted into the 3D bioprinter. Subsequently, the MAT and fibrin glue were printed on the inside of the PCL scaffold in order, using extrusion-based bioprinting and a layer-by-layer approach to create a MAT niche. After hardening, the inner part of the MAT niche was elevated using forceps, except the PCL scaffold, and applied to the wound bed after wound cleaning (Figure 2). Lastly, the wound was covered with a sterile silicone wound dressing (Mepitel^®^; Mölnlycke Health Care, Gothenburg, Sweden). The patients were monitored with weekly dressing changes. 

#### 2.1.3. Control Group: Standard Wound Care

Subjects randomized in the control group underwent standard wound care; weekly follow-up was performed as in the treatment group. 

All patients in both groups with ulcers on weight-bearing sites or sites otherwise subjected to pressure when wearing shoes had pressure offloaded using foam dressings with a hole on the ulcer site and footwear with cushioning insoles. 

### 2.2. Evaluation

The primary efficacy criterion was the mean percentage change in the wound area at the fourth week. The secondary efficacy criterion was the percentage change of patients who achieved complete wound closure within the 16 week study period. Complete wound closure was defined as a completely epithelialized state in the absence of any discharge, which allowed the patient to shower. The wound area was determined using digital photometry at each visit. Safety was also assessed by evaluating adverse event reports throughout the trial. 

### 2.3. Statistical analysis

Statistical comparisons for healing rates were performed using Pearson’s chi-squared test. The time required to achieve complete epithelialization for each patient was assessed using the Wilcoxon signed rank test. The median time to closure was estimated using the Kaplan–Meier method. Data were expressed as the mean ± standard deviation. A *p*-value < 0.05 was considered statistically significant. All statistical analyses were performed using the SPSS version 20.0 for Windows (SPSS Inc., Chicago, IL, USA).

## 3. Results

### 3.1. Mean Percentage of Wound Area Reduction from Baseline to 4 Week Visit 

The wound size reduced by 4.3 ± 1.0 cm^2^ in the treatment group and by 2.0 ± 1.1 cm^2^ in the control group. The reduction rates were 77.1% ± 4.9% and 45.7% ± 15.6% in the treatment and control groups, respectively (Figure 3).

### 3.2. Percentage of Patients Who Achieved Complete Wound Closure within 16 Weeks

Complete wound healing was achieved after 16 weeks in eight out of 10 patients (80%) and three out of six patients (50%) in the treatment and control groups, respectively (Figure 4, Figure 5 and Figure 6).

The time required to achieve complete healing was 10.2 ± 1.4 weeks in the treatment group and 13.3 ± 1.9 weeks in the control group (Figure 7).

### 3.3. Adverse Events

Adverse events occurred in one patient in the control group; the patient reported a hypoglycemia symptom owing to maladjusted diet and insulin concentration. Although the adverse event occurring in the patient allocated in the control group was not directly related to the present clinical trial, the patient was withdrawn to receive proper treatment for their hypoglycemic symptoms. Otherwise, no complications, including infection, hemorrhage, and allergic reactions, were observed in either group after the respective treatment.

## 4. Discussion

The patients who received the MAT niche treatment had better results than the patients who received standard wound care, in terms of reduction in the wound area after the 4 week treatment period (*p* = 0.043). The wound reduction rate during the 4 week treatment period was also higher in the MAT niche treatment group than in the standard wound care group (77.1% vs. 45.7%; *p* = 0.055). Here, change in the wound area during the 4 week treatment period has been widely adopted as a clinical indicator to predict healing [13,14,15], and the results in the present study were favorable. Moreover, in the MAT niche treatment group, the percentage of patients who achieved complete wound closure after 16 weeks was higher, and the time required to achieve complete healing was shorter than those in the standard wound care group, although the differences were not statistically significant (*p* = 0.299, Fisher’s exact test; *p* = 0.187, the log rank test using the Kaplan–Meier method, respectively). The weakness in statistical power in the current study may be ascribed to the fact that the total number of patients enrolled was small. To attain a high statistical power, a clinical study that includes a larger number of participants is needed. In addition, the safety and tolerability results were acceptable; no adverse events occurred related to the treatment. Therefore, our findings suggest that the MAT niche treatment may pose clinical benefits to patients.

Diabetes affects the normal healing process at the molecular level. Several key factors, including reduced cell proliferation and migration, slower growth factor production, and an abnormal ECM, contribute to impaired wound healing in patients with diabetes [16,17,18,19]. To overcome these obstacles, MAT niche treatment might be a promising therapeutic alternative in that it is composed of cellular components such as ASCs [3,20,21,22,23,24,25,26,27,28], ECMs [29,30,31,32,33,34], and various cytokines [4,35], which are fundamental factors in wound healing. 

Previous studies have confirmed that the primary role of adipose tissue in promoting rejuvenation is played by ASCs in stromal vascular fraction (SVF) cells. Several mechanical processing procedures, including centrifugation, mechanical chopping, shredding, pureeing, and mincing, have been developed to obtain ASCs from adipose tissue without collagenase-mediated digestion [36,37,38]. These methods are speculated to condense tissue and ASCs by mechanically disrupting mature adipocytes and their oil-containing vesicles. Moreover, these preparations, which contain a high density of ASCs, show considerable therapeutic potential as regenerative medicine [39]. In 2016, Ceserani et al. reported that mechanically derived MAT retains anti-inflammatory and angiogenic properties even better than those of the cultured adipose-derived mesenchymal stem cell content in vitro [10]. Nava et al. also demonstrated the long-lasting anti-inflammatory activity of human MAT in vivo in 2019 [11]. In addition, the application of MAT in vivo has shown some beneficial effects of releasing angiogenic and anti-inflammatory molecules [10,40]. Considering the above beneficial effects, MAT is considered to be a promising and safe option to overcome the aforementioned complex regulatory issues associated with enzymatic treatment or cell expansion. This study is the first to demonstrate that MAT may offer the possibility of an effective treatment for diabetic ulcers, particularly utilizing the 3D bioprinting technology to customize an ECM niche scaffold according to the patient’s wound structure. 

By adding the fibrin glue to the MAT niche, which was fabricated using 3D bioprinting technology, it could be applied, maintained, and fixed to the wound as an optimized form, fitted to the shape and depth of the wound and the convolution of the wound bed. In addition, the regenerative effect of thrombin or fibrinogen, which constitute the fibrin glue, may also contribute to the healing potential of the MAT niche treatment [41,42,43,44,45,46,47].

Collectively, substances, including (1) various cell components such as ASCs, (2) ECM, (3) various cytokines or healing-related peptides present in the autologous fat, and (4) fibrin glue may have a positive effect on the wound-healing process in the present study. 

Recent advances in 3D bioprinting technology have prompted our group to create a MAT niche to fit DFUs. The MAT niche may provide an efficient environment during cell proliferation and differentiation; this technique guarantees less invasiveness with no direct route toward the intra-circulation. The structure of a MAT niche could also facilitate the volume replacing effect that helps shorten the healing period by inducing stem0cell homing or paracrine signaling effects. 

The MAT niche autograft could detour the limitation of SVF transplantation in that enzymatic digestion using collagenase is not necessary; thus, regulatory issues could be avoided. The use of minimally manipulated autologous adipose tissue complies with ethical laws. Immunological rejection could be avoided without heterologous and/or allogeneic material. Moreover, grafting the MAT niche to the wound defect avoids possible complications related to endovascular delivery into the systematic circulation. 

This study had limitations. It included a small cohort. Further studies with larger cohorts are necessary to demonstrate the statistically significant superiority of the MAT niche autograft over conventional treatment. In addition, wound evaluation was performed in a nonblinded manner. The patients and the clinician both knew the assignment of the MAT niche-treated or control group. To determine the ultimate value of the described method, a further blinded study with a larger sample size may be necessary. 

Importantly, the present study does not support the use of this therapy to treat foot ulcers in patients with diabetes because of the small number of patients included and the absence of statistical significance. However, the results of this pilot preliminary study are promising, as MAT niche autografts may offer the possibility of a simple and effective treatment for diabetic ulcers. Therefore, further follow-up studies with a larger number of patients are required to validate our findings.

## Figures and Tables

**Figure 1 jcm-11-05887-f001:**
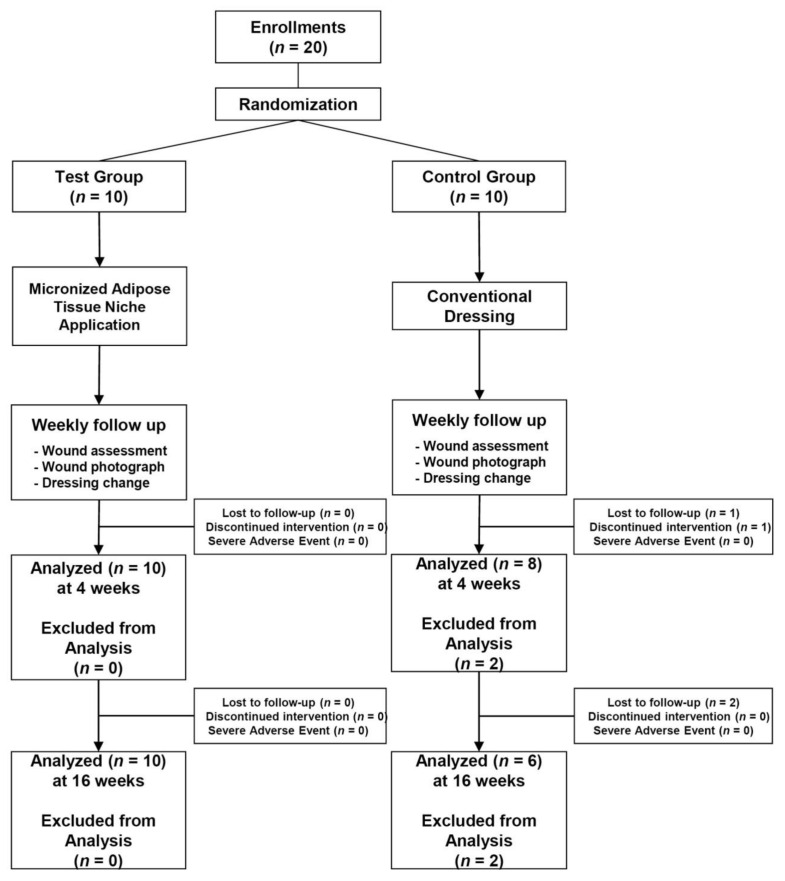
CONSORT diagram showing the flow of patients included in this study.

**Figure 2 jcm-11-05887-f002:**
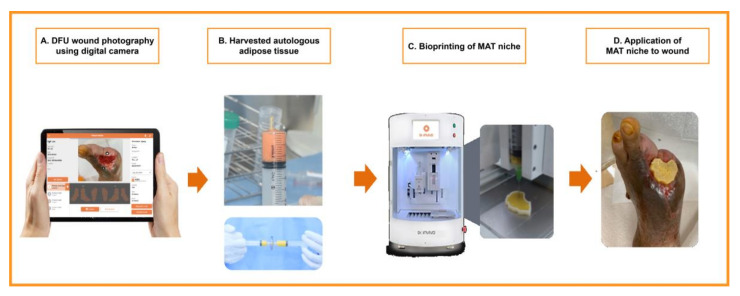
A diagram showing the flow of patients in the treatment group in this study.

**Figure 3 jcm-11-05887-f003:**
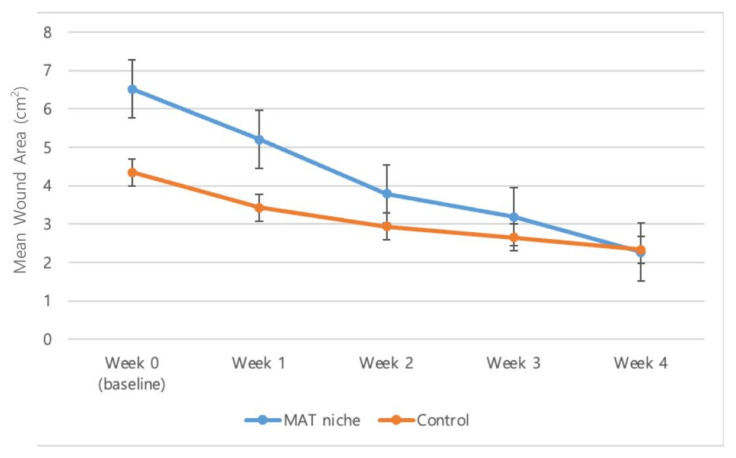
The mean wound area from baseline to the 4 week visit. The wound size was reduced by 4.3 ± 1.0 cm^2^ (mean ± SD) and 2.0 ± 1.1 cm^2^ in the treatment and control groups, respectively. The reduction rates were 77.1% ± 4.9% and 45.7% ± 15.6% in the treatment and control groups, respectively.

**Figure 4 jcm-11-05887-f004:**
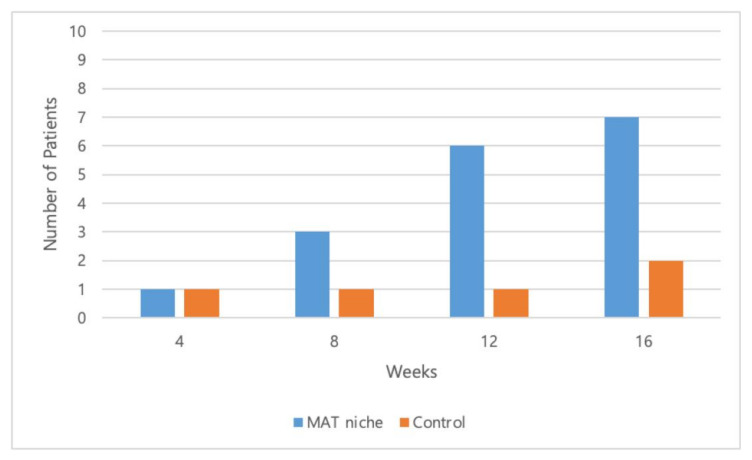
Total number of patients with completely closed wounds at 4, 8, 12, and 16 weeks. Complete wound closure was achieved in 10%, 40%, 70%, and 80% of patients in the treatment group and in 16.7%, 16.7%, 16.7%, and 50.0% of patients in the control group at 4, 8, 12, and 16 weeks, respectively.

**Figure 5 jcm-11-05887-f005:**
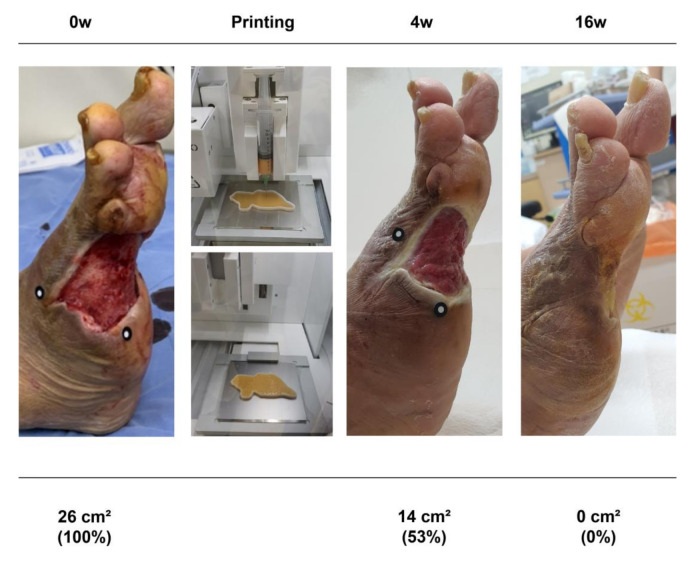
A 70 year old man with diabetes mellitus had a nonhealing ulcer on his right foot for 8 weeks. A micronized adipose tissue niche was applied to the wound.

**Figure 6 jcm-11-05887-f006:**
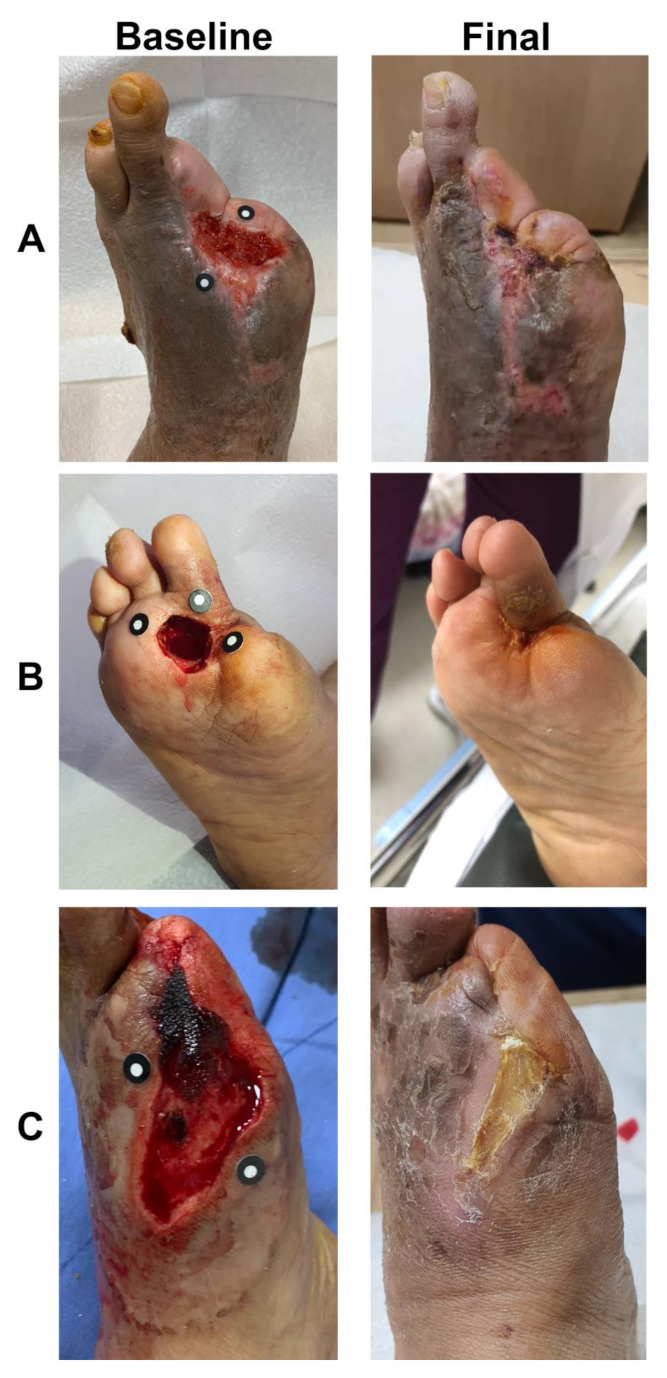
Three representative examples of micronized adipose tissue niche grafts of diabetic foot ulcers. Baseline: before treatment. Final: after treatment at the first closure at 6, 10, and 11 weeks in Patients (**A**), (**B**), and (**C**), respectively).

**Figure 7 jcm-11-05887-f007:**
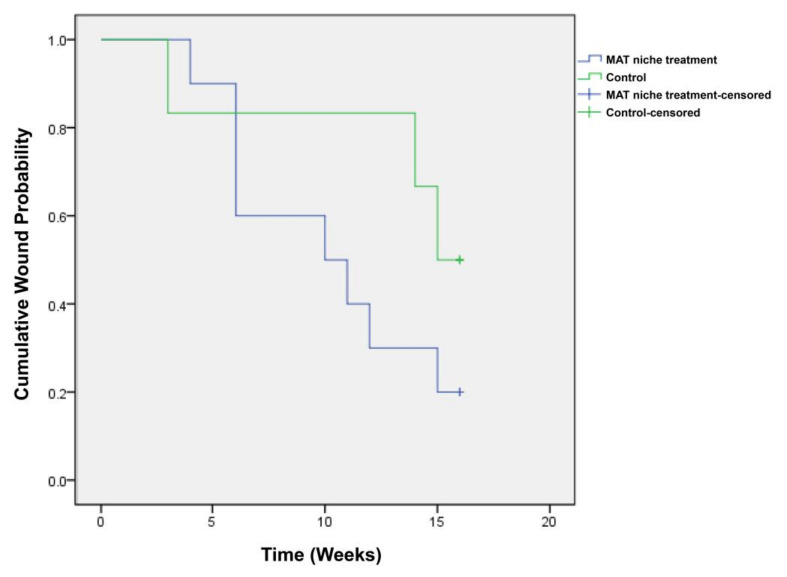
Kaplan–Meier diagram showing results of the time to wound closure. The Kaplan–Meier median times to complete closure were 10.2 ± 1.4 and 13.3 ± 1.9 weeks in the treatment and control groups, respectively.

**Table 1 jcm-11-05887-t001:** Patient characteristics in the micronized adipose tissue (MAT) niche treatment and control groups.

	MAT Niche (*n* = 10)	Control (*n* = 10)	*p*-Value
Age (years)	59.3 ± 9.92	56.9 ± 11.39	0.68 ^a^
Sex (%)			1.00 ^b^
Men	8 (80)	9 (90)	
Women	2 (20)	1 (10)	
Type of Diabetes (%)			1.00 ^b^
Type 1	0 (0)	1 (10)	
Type 2	10 (100)	9 (90)	
Ulcer location (%)			0.44 ^b^
Forefoot	9 (90)	6 (60)	
Midfoot	0 (0)	2 (20)	
Hindfoot	1 (10)	1 (10)	
Above ankle	0 (0)	1 (10)	
Wagner grade (%)			1.00 ^b^
Grade 1	7 (70)	8 (80)	
Grade 2	3 (30)	2 (20)	
Diabetic peripheral neuropathy	10 (100)	10 (100)	1.00 ^b^
Glycated hemoglobin, mean (SD), %	7.14 ± 1.09	7.17 ± 0.88	0.80 ^a^
Hemoglobin, mean (SD), g/dL	11.41 ± 0.90	11.40 ± 1.65	0.97 ^a^
Albumin, mean (SD), g/dL	3.79 ± 0.45	4.10 ± 0.20	0.17 ^a^
Creatinine (SD), mg/dL	3.32 ± 2.87	2.67 ± 2.82	0.80 ^a^
Estimated glomerular filtration rate (SD), mL/min	49.35 ± 44.53	72.20 ± 72.85	1.00 ^a^
C-reactive protein, mean (SD), mg/L	3.17 ± 3.12	6.55 ± 6.35	0.48 ^a^
Erythrocyte sedimentation rate, mean (SD), mm/h	57.40 ± 33.35	54.50 ± 25.34	0.80 ^a^
White blood cells, mean (SD), ×10^3^/μL	6.36 ± 1.80	7.74 ± 3.12	0.19 ^a^
Ulcer size, mean (SD), cm^2^	5.66 ± 4.74	4.66 ± 6.86	0.09 ^a^

^a^ Mann—Whitney test, with *p* < 0.05 considered statistically significant. ^b^ Pearson’s chi-squared test, with *p* < 0.05 considered statistically significant.

## Data Availability

Not applicable.

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
