# Peer review of "A Pilot Study Comparing a Micronized Adipose Tissue Niche versus Standard Wound Care for Treatment of Neuropathic Diabetic Foot Ulcers"

_jcm, 2022, doi:10.3390/jcm11195887_

Round 1
Reviewer 1 Report
The paper is scientific interesting.
However, there are methodological concerns, such as being not-blinded, and, though randomized, the wound area at baseline was significantly greater in the intervention group compared to the control group. Furthermore, the number of patients in the control group decreases (to 6? or 8? during the study. The p values of comparisons between the two groups are mostly not statistically significant or around 0.05. Given the difference in baseline wound area between the groups, this study can only give an impression of tendencies.
How was "did not show signs of clinical healing 6 weeks prior to inclusion" defined
Please describe the standard treatment of the ulcers in both groups: were the ulcer debribed at each visit? or?
Author Response
Reviewer #1:
Comments to the Author
The paper is scientific interesting. However, there are methodological concerns, such as being not-blinded, and, though randomized, the wound area at baseline was significantly greater in the intervention group compared to the control group. Furthermore, the number of patients in the control group decreases (to 6? or 8?) during the study. The p values of comparisons between the two groups are mostly not statistically significant or around 0.05. Given the difference in baseline wound area between the groups, this study can only give an impression of tendencies.
(Response) We thank the reviewer for their comments. We appreciate the valuable time and effort that the reviewer has dedicated towards reviewing our manuscript. We agree with the reviewer's comments regarding the methodological concerns in the present study. Certainly, being blinded with a well-controlled baseline comparison would strengthen our study. However, the study was conducted as a pilot study to determine whether this novel procedure can accelerate wound healing in diabetic foot ulcers. Due to the setting of the pilot study that accompanies the surgical procedure of fat harvesting with the processing of adipose tissue micronization, a double-blinded design was not feasible. The decrease in the number of patients included in the study from 8 to 6 was also inevitable because the study abided with the informed consent rules under the IRB protocol. As we have previously documented in the Materials and Methods section, two patients in the control group were excluded in the primary evaluation period (one patient dropped out because of hypoglycemia from accompanying diabetes, and the other chose to withdraw). In the secondary evaluation period, which was conducted only for patients who could be followed up until 16 weeks post-treatment, a total of 16 patients were included. We agree that the total number of patients was too small to be presented as an RCT study; thus, we have removed the description regarding RCTs throughout the revised manuscript. However, to the best of our knowledge, no clinical studies have reported the effects of MAT on diabetic wound healing, and this study is clinically significant as a pilot study in that this novel technique using MAT niche autografts may offer the possibility of treating diabetic ulcers. We have detailed the current study's limitations in the Discussion section on page 11. Based on the results of the present pilot study, we plan to conduct follow-up investigations with well-controlled baseline wound areas in both groups consisting of more patients under randomized clinical settings.
How was "did not show signs of clinical healing 6 weeks prior to inclusion" defined
(Response) The phrase in the inclusion criteria, “presence of lower extremity ulcers that did not show signs of healing in the 6 weeks prior to inclusion,” refers to the ulcers that show no signs of decrement of wound dimension with regard to granulation formation, contraction, and epithelialization. Generally, wounds heal by granulation formation (fibrosis), contraction, and epithelialization (SK Han, Innovations and Advances in Wound Healing, Springer, 2015). Thus, if the wound did not show signs of wound epithelialization, granulation formation, and wound contraction for 6 weeks prior to inclusion in the study, it could be regarded as a chronic ulcer that did not heal easily. This information has been added in the revised manuscript on page 2.
Please describe the standard treatment of the ulcers in both groups: were the ulcer debrided at each visit? or?
(Response) The standard treatment protocol for diabetic foot ulcers in both groups has been added to the Methods section of the revised manuscript. The ulcers that need debridement were debrided at each visit before enrollment in the present study as part of the treatment protocol at our diabetic wound center. All ulcers were subjected to sharp debridement to remove necrotic or hyperkeratinized tissue and control infection prior to study initiation. After the initiation of the present study, the ulcers did not undergo debridement in either group, which might hinder the interpretation of the results. This information has been detailed in the Materials and Methods section (2.1.1. Brief management protocol of diabetic foot ulcers) of the revised manuscript on page 4.

Reviewer 2 Report
Major comment
The authors of the present study describe the treatment of patients with advanced diabetic foot lesions, probably of neuropathic origin, with fat microclusters embedded in custom-made scaffolds of polycaprolactone using a bioprinting technique. The goal of the study is to determine whether this procedure can accelerate wound healing. This question is highly topical and of great clinical importance. The methods used are relatively new and innovative. The authors conclude that treating wounds in this way results in significantly faster healing.
The study is presented as a randomized controlled trial (RCT) and evaluated accordingly. In purely formal terms, the study is indeed an RCT, but due to the far too small number of patients treated, statistical evaluation in the sense of an RCT is not possible. Therefore, the results of the present study cannot be interpreted as proof of the efficacy of the therapy and certainly not of its safety, but only as a preliminary indication of this.
Minor comments
Title: The title should be changed to: "A pilot study comparing a micronized adipose tissue niche versus standard wound care for treatment of neuropathic diabetic foot ulcers".
Introduction: All literature on previously published papers on the treatment of wounds with fat grafting is not mentioned. There are at least 3 reviews on this topic (e.g., Int Wound J, 2019, 16:275).
Statistical methods: The study should not be presented as an RCT. An important reason for conducting an RCT is to avoid selection bias in assigning patients to treatment and control groups with the goal of establishing comparability between the two groups. The number of patients required for this purpose is calculated before the start of the study. This calculation of the sample size is completely missing in the present work. Such a calculation would probably have resulted in a sample size of about 50 (standardized difference for wound area 1, power 0.8, significance level 0.95). The fact that therapy and control groups are not comparable in the study can easily be seen from the clearly different baseline wound areas.
Patient characteristics: The examination for peripheral neuropathy is missing.
Results: The results should be presented purely descriptively without the use of statistical tests.
Discussion: The results should be discussed from the point of view that they may provide evidence for efficacy of the presented therapy. The present discussion can be significantly shortened. The pathophysiological mechanisms behind the possible success of the therapy are very interesting, but not the main subject of the study.
Conficts of interest: It is surprising that Ji-Won Son has no conflict of interest, although he is employed by the company sponsoring the study.
Author Response
Reviewer #2:
Comments to the Author
Major comment
The authors of the present study describe the treatment of patients with advanced diabetic foot lesions, probably of neuropathic origin, with fat microclusters embedded in custom-made scaffolds of polycaprolactone using a bioprinting technique. The goal of the study is to determine whether this procedure can accelerate wound healing. This question is highly topical and of great clinical importance. The methods used are relatively new and innovative. The authors conclude that treating wounds in this way results in significantly faster healing.
(Response) We thank the reviewer for their comment. We appreciate the valuable time and effort that the reviewer has dedicated towards reviewing our manuscript.
The study is presented as a randomized controlled trial (RCT) and evaluated accordingly. In purely formal terms, the study is indeed an RCT, but due to the far too small number of patients treated, statistical evaluation in the sense of an RCT is not possible. Therefore, the results of the present study cannot be interpreted as proof of the efficacy of the therapy and certainly not of its safety, but only as a preliminary indication of this.
(Response) We thank the reviewer for their comment. As we have documented in the Discussion section, the current study is limited on account of the total number of patients treated being too small. We also agree with the reviewer’s comment that a statistical evaluation as is expected for an RCT was not possible in the present study setting owing to the limited number of subjects, although this study was indeed an RCT in purely formal terms. Accordingly, we have deleted the expressions that emphasize that the study was conducted as an RCT in the title and throughout the manuscript and have changed the title in the revised version of the document. However, as the reviewer has pointed out, the present study demonstrates the possibilities of clinical application of an innovative bioprinting technique with micronized adipose tissue for diabetic foot ulcers as a pilot study, which would benefit the readers of this esteemed journal. This information has been added to the Discussion section of the revised manuscript.
Minor comments
Title: The title should be changed to: "A pilot study comparing a micronized adipose tissue niche versus standard wound care for treatment of neuropathic diabetic foot ulcers".
(Response) In accordance with the reviewer’s suggestion, we have changed the title of the manuscript.
Introduction: All literature on previously published papers on the treatment of wounds with fat grafting is not mentioned. There are at least 3 reviews on this topic (e.g., Int Wound J, 2019, 16:275).
(Response) In accordance with the reviewer’s suggestion, we have added the three recent reviews on this topic (References 7-9), including the article by Smith et al. in Int Wound J in the Introduction section of the revised manuscript on page 1.
Statistical methods: The study should not be presented as an RCT. An important reason for conducting an RCT is to avoid selection bias in assigning patients to treatment and control groups with the goal of establishing comparability between the two groups. The number of patients required for this purpose is calculated before the start of the study. This calculation of the sample size is completely missing in the present work. Such a calculation would probably have resulted in a sample size of about 50 (standardized difference for wound area 1, power 0.8, significance level 0.95). The fact that therapy and control groups are not comparable in the study can easily be seen from the clearly different baseline wound areas.
(Response) We thank the reviewer for raising this point. We agree that the present study should not be presented as an RCT; therefore, we have deleted the related terminology throughout the revised manuscript. The present pilot study is clinically significant in that it has demonstrated the possibilities of treating diabetic foot ulcers with micronized adipose tissue compared with conventional treatment, which has never been previously demonstrated. Thus, based on the present study, we have planned to conduct a follow-up study under a randomized clinical trial with a sufficient sample size in the near future.
Patient characteristics: The examination for peripheral neuropathy is missing.
(Response) We thank the reviewer for their insightful comments, which have helped us improve our manuscript significantly. Based on the reviewer’s suggestion, we retrospectively reviewed the data on peripheral neuropathy of all the enrolled patients, and the EMG/NCVs results of both groups demonstrated that all the patients enrolled have peripheral neuropathy; therefore, we have added the clinical characteristics regarding the examination for peripheral neuropathy to the revised version of Table 1 on page 4.
Results: The results should be presented purely descriptively without the use of statistical tests.
(Response) Per the reviewer’s comments, we have presented the results purely descriptively without the use of statistical tests in the Results section of the revised manuscript (pages 6-9).
Discussion: The results should be discussed from the point of view that they may provide evidence for efficacy of the presented therapy. The present discussion can be significantly shortened. The pathophysiological mechanisms behind the possible success of the therapy are very interesting, but not the main subject of the study.
(Response) Thank you for this comment. As per the reviewer’s comments, we have significantly shortened the previous Discussion section of 4 paragraphs, which covered the pathophysiological mechanisms of the therapy, into one paragraph in the revised manuscript on page 10.
Conflicts of interest: It is surprising that Ji-Won Son has no conflict of interest, although he is employed by the company sponsoring the study.
(Response) We thank the reviewer for their comment. As the reviewer has rightly pointed out, the fourth author of the present study, Jeehee Kim, is currently employed by the company sponsoring the study, and it should be presented clearly in the conflicts of interest section. Thus, we have included this point in the Conflict of Interest section in the revised manuscript.

Round 2
Reviewer 1 Report
The comments given by the reviewers have been adressed in the revised manuscript, - however the abstract could to be revised accordingly, as well
Author Response
(Response) We have revised the abstract as you suggested (page 1).

Reviewer 2 Report
The authors have given sufficient consideration to the reviewers' comments. In my opinion, the paper can be accepted for publication after this revision.
Author Response
(Response) We thank the reviewer for their comment. We appreciate the valuable time and effort that the reviewer has dedicated towards reviewing our manuscript.
